# Efficacy of the Second COVID-19 Vaccine Booster Dose in the Elderly

**DOI:** 10.3390/vaccines11020213

**Published:** 2023-01-18

**Authors:** Camilla Mattiuzzi, Giuseppe Lippi

**Affiliations:** 1Service of Clinical Governance, Provincial Agency for Social and Sanitary Services, 38123 Trento, Italy; 2Section of Clinical Biochemistry and School of Medicine, University of Verona, 37134 Verona, Italy

**Keywords:** COVID-19, SARS-CoV-2, vaccination, booster, elderly

## Abstract

Background: We analyzed coronavirus disease 2019 (COVID-19) vaccine efficacy in older persons who received the second booster compared to unvaccinated people and those receiving only a single COVID-19 vaccine booster. Methods: We collected information on vaccine efficacy from the ongoing Italian nationwide COVID-19 vaccination campaign in subjects aged 80 years or older from official data published by the Italian National Institute of Health. Results: The second vaccine booster maintained high effectiveness against adverse COVID-19 outcomes such as hospitalization, intensive care unit admission and death (i.e., between 77 and 86%), and also showed around 10% higher efficacy than the single booster. Nonetheless, the efficacy of the second vaccine booster declined over time, decreasing by 33–46% when assessed at >120 days from administration. Conclusions: The results of our ad interim analysis of the ongoing Italian nationwide COVID-19 vaccination campaign suggest that regular boosting with COVID-19 vaccines may be advisable in older persons.

## 1. Introduction

Coronavirus disease 2019 (COVID-19), a life-threatening infectious pathology first identified in the Chinese town of Wuhan in 2019, has since spread all over the world, becoming one of the most lethal infectious diseases throughout the known human history [1]. Although widespread COVID-19 vaccination combined with gradual mitigation of virus aggressiveness have indeed contributed to reducing the clinical and social implications of severe acute respiratory syndrome coronavirus 2 (SARS-CoV-2) infection over time, reliable evidence suggests that older individuals remain at an exceptionally enhanced risk of developing unfavorable COVID-19 progression, as recently highlighted by Maynou et al. [2], who emphasized that the risk of COVID-19-related death is considerably high in this part of the population, irrespective of the frailty status.

We have previously emphasized that the risk of COVID-19-related hospitalization, intensive care unit (ICU) admission and death was 80% lower in older Italian persons who received the first COVID-19 vaccine booster compared to those who earlier completed the primary vaccination cycle [3]. Since both the natural and vaccine-elicited immunities gradually wane over time, compounded by the fact that new and highly mutated SARS-CoV-2 variants are continuously emerging and spreading all around the world, the administration of additional vaccine booster doses (i.e., the fourth or even fifth) has commenced in several countries worldwide, especially directed to older people and those with various causes of immune system impairment. Nonetheless, reliable correlates of vaccine protection are still lacking, whilst the clinical evidence emerged so far from large real-world studies that this strategy would be clinically effective to avert infections, COVID-19-related hospitalizations and deaths remains scarce. Many national governments and health institutes are hence questioning whether this practice shall be continued, while the willingness to undergo administration of further COVID-19 vaccine boosters has concomitantly waned over time in the general worldwide population. Thus, since the administration of a second COVID-19 booster dose of mRNA-based vaccines (i.e., Pfizer/BioNTech Comirnaty and Moderna Spikevax) was initiated in Italy and April 2022 in persons aged 80 years or older, we provide here updated statistics on its efficacy in this more vulnerable part of the general population.

## 2. Materials and Methods

Our analysis was based on official data of the COVID-19 national integrated surveillance program, regularly updated by the Italian National Institute of Health (Istituto Superiore di Sanità, ISS (last available update, 18 November 2022) [4]. The official ISS bulletin contains updated nationwide information on the cumulative burden of COVID-19-related hospitalizations, ICU admissions and deaths. The data published by the ISS originates from the COVID-19 integrated Italian surveillance system and are analyzed by integrating microbiological and epidemiological data provided by regions and autonomous provinces and by COVID-19 National Reference Laboratories. Specifically, all regions and autonomous provinces digitally transmit daily information to the ISS concerning all individuals undergoing vaccination and those diagnosed with SARS-CoV-2 infection (i.e., all subjects with laboratory diagnosis of COVID-19, regardless of clinical signs and symptoms). The ISS has created a dedicated IT platform, interfaced with a specific database, which allows data to be collected through a web interface remotely connected to the platform. All COVID-19 cases diagnosed by regional reference laboratories and other laboratories within the diagnostic network participate to the surveillance program. Information on pre-existing pathologies and clinical conditions are also entered within the digital platform. The official data of the ISS bulletin were retrieved and transcribed into an Excel worksheet (Microsoft Excel; Microsoft, Redmond, WA, USA), whilst vaccine efficacy over time was estimated by calculating the odds ratio (OR) and 95% confidence interval (95%CI) for each of the three major endpoints (COVID-19-related hospitalizations, ICU admissions and deaths) with MedCalc (Version 20.015; MedCalc Software Ltd., Ostend, Belgium).

The study was performed in accordance with the Declaration of Helsinki, under all relevant terms of the local legislation. The research was based on public ISS data [4], so that ethical committee approval and informed consent were both unnecessary.

## 3. Results

According to the official statistics of the Italian National Institute of Health, the number of older persons (i.e., aged ≥80 years) who were still unvaccinated, who had received the first COVID-19 vaccine booster and who had received the second COVID-19 vaccine booster at 18 November 2022 were 140,164, 2,542,606 and 1,536,464, respectively (this statistic does not consider the time passed since vaccination). The results of our analysis are summarized in Table 1 and Figure 1.

As specifically shown in Figure 1, the rate of COVID-19-related hospitalizations, ICU admissions and deaths progressively declined from people who had not undergone COVID-19 vaccination, to those who had received a single COVID-19 vaccine booster, and finally to those who received the second vaccine booster.

The risk of COVID-19-related hospitalization, ICU admission and death was also nearly 10% lower in those who received the second COVID-19 vaccine booster dose compared to those who only received the first booster, with this difference achieving statistical significance for COVID-19-related hospitalizations. A sub-analysis of older people who received the second COVID-19 vaccine booster also showed that vaccine efficacy declined over time, in that the risk of COVID-19-related hospitalizations, ICU admissions and deaths increased by 33%, 47% and 44% in older subjects who had received the second COVID-19 vaccine booster after 120 days compared to those who instead received the second vaccine booster within 120 days, respectively (Table 2).

## 4. Discussion

Several lines of evidence now unquestionably confirm that vaccination is the most important, practical and globally acceptable strategy to face the ongoing COVID-19 pandemic. A vast array of clinical studies have confirmed that COVID-19 vaccines are highly effective, even more so than those used for influenza vaccination (i.e., 80–90% vs. 50–60% efficacy against laboratory-confirmed symptomatic infections) [5], and relatively safe, since the burden of serous side-effects is orders of magnitude lower than the organic injuries caused by COVID-19 [6]. Besides the clinically favorable balance between efficacy and side effects, a recent systematic literature review also concluded that universal administration of COVID-19 vaccine is a cost-effective and cost-saving strategy for attenuating viral spread and averting the many adverse clinical consequences caused by COVID-19 illness [7]. Among the general population, older individuals are those more vulnerable to COVID-19 illness, in that the risk of unfavorable clinical progression, hospitalization, ICU admission and death is magnified compared to younger people [8]. Although the favorable effects of primary COVID-19 vaccination on the risk of infection and/or developing severe/critical illness have been definitely ascertained, the use of booster doses remains controversial, and the number of larger studies, on a nationwide scale, is still limited, thus leaving residual doubts as to whether and when vaccine boosters would need to be administered [9]. The main reason underneath this open debate is that vaccine boosters may not ultimately be cost-effective for enhancing protection and could hence divert more-needed doses away from those who are still waiting to receive the primary vaccination cycle, especially in countries were vaccine availability is lower. This is in keeping with a statement of the World Health Organization (WHO), which recently affirmed that the risk of wasting many more resources in boosting people already efficiently protected against the likelihood of adverse clinical progression is a tangible problem [9]. Thus, more large, nationwide studies are needed to definitely prove that repeated administration of vaccine boosters would be an efficient strategy against COVID-19, especially in frail and other more vulnerable parts of the general population.

The results of our ad interim analysis of the ongoing Italian nationwide COVID-19 vaccination campaign [10] provide clear evidence that the administration of a second COVID-19 vaccine booster dose may be substantially advantageous in older persons. In particular, we showed that vaccine efficacy against all the most important endpoints (i.e., COVID-19-related hospitalizations, ICU admissions and deaths) not only could be maintained at exceptionally high levels (i.e., between 77 and 86%) in those who received the second COVID-19 vaccine booster, but was also constantly higher (by around 11–13%, in absolute terms) compared to that administered only the first, single booster. These results are well aligned with those very recently published by Kislaya et al., who also showed that the second COVID-19 vaccine booster in older Portuguese individuals (i.e., aged ≥80 years) conferred 81–82% protection against the risk of hospitalization or death compared to a vaccine efficacy of around 63–64% in those who only received the first COVID-19 vaccine booster [11]. Unlike this report, however, we also found that the efficacy of the second COVID-19 vaccine booster gradually declined over time, with a trend mirroring that already seen for primary vaccination and for the first vaccine booster [12]. This is rather understandable, considering that both humoral and cellular immunity against SARS-CoV-2 tend to decline over time, even after administration of vaccine boosters, as clearly demonstrated by many recent studies [13,14,15]. Importantly, cellular immunity and immunological memory seem to wane slower and/or at lower extent compared to humoral immunity, and this evidence provides a reasonable explanation for the fact that vaccine efficacy after the second COVID-19 vaccine booster is not exceptionally superior to that conferred by the single booster [15]. Unfortunately, there is no simultaneous information on younger and elderly people that would enable a comparison of the efficacy of the second vaccine booster, since its administration only commenced in people aged <60 years at the end of 2022, when different SARS-CoV-2 variants (i.e., BQ.1 and XBB.1 vs. BA.5) became endemic, thus making the comparison of efficacy of the second vaccine booster potentially misleading.

## 5. Conclusions

The results of our ad interim analysis of the ongoing Italian nationwide COVID-19 vaccination campaign suggest that repeated boosting with COVID-19 vaccines may be advisable in the elderly, for clinical, social and even economic reasons. To this end, the cost-effectiveness of administering a second COVID-19 vaccine booster dose has been recently highlighted by Mungmunpuntipantip and Wiwanitkit [16], who concluded that the cost per effectiveness of this practice is 66% higher than limiting vaccine coverage to administration of the first vaccine booster (i.e., USD 28.68 vs. USD 47.56). We also emphasize that further and future boosters should preferably be administered using COVID-19 bivalent vaccines, since their efficacy in neutralizing emerging and highly mutated SARS-CoV-2 sublineages seems higher than that of the previous monovalent formulations [17,18,19,20]. The safety profile of the fourth dose also seems reassuring. Akaishi et al. [21] compared the burden of acute side-effects recorded after receiving four doses of mRNA-based COVID-19 vaccines, and reported that the incidence rates after the third and fourth doses were significantly lower than those seen after the first and second administrations. The authors also found that the risk of developing vasovagal syncope/presyncope was lower after the fourth compared to the third vaccine dose (0.02 vs. 0.09%: *p* < 0.001), as was the likelihood of developing post-vaccine allergic reactions (0.01 vs. 0.03%; *p* = 0.010). Thus, apparently, the administration of a fourth COVID-19 vaccine dose does not raise more safety issues than the third dose. Further studies would instead be needed to define whether a personalized approach to vaccination, based on surrogate measures of cellular (e.g., interferon-gamma release assays) and humoral (e.g., anti-SARS-CoV-2 neutralizing antibodies) immunity, could help optimize COVID-19 vaccines administration even in elderly and/or fragile populations.

## Figures and Tables

**Figure 1 vaccines-11-00213-f001:**
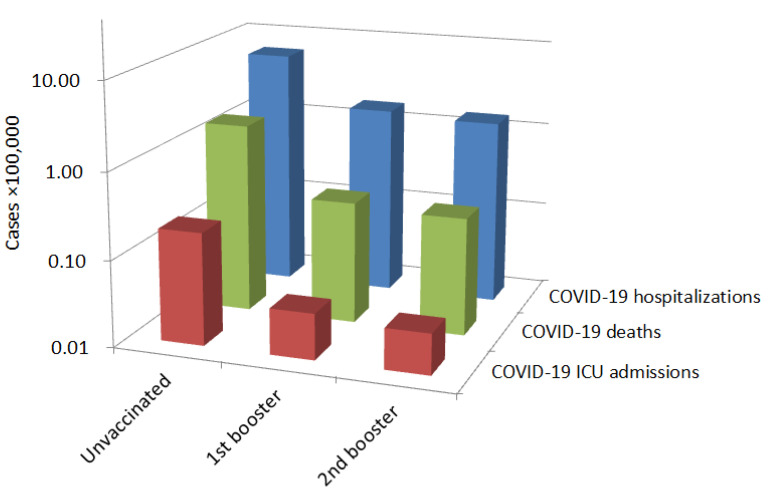
Cases per 100,000 of coronavirus disease 2019 (COVID-19)-related hospitalizations, intensive care unit (ICU) admissions and deaths within the nationwide COVID-19 vaccination campaign in older Italian persons (i.e., aged >80 years).

**Table 1 vaccines-11-00213-t001:** Efficacy of the first and second coronavirus disease 2019 (COVID-19) vaccine booster doses in older Italian persons (i.e., aged ≥80 years).

Endpoint	Unvaccinated	2nd Vaccine Booster
OR (95%CI) COVID-19-related hospitalizations		
1st vaccine booster	0.26 (95% CI, 0.24–0.28; *p* < 0.001)	0.89 (95% CI, 0.84–0.94; *p* < 0.001)
2nd vaccine booster	0.23 (95% CI, 0.21–0.25; *p* < 0.001)	-
OR (95%CI) COVID-19-related ICU admissions		
1st vaccine booster	0.17 (95% CI, 0.11–0.26; *p* < 0.001)	0.87 (95% CI, 0.61–1.24; *p* = 0.443)
2nd vaccine booster	0.15 (95% CI, 0.09–0.24; *p* < 0.001)	-
OR (95%CI) COVID-19-related deaths		
1st vaccine booster	0.16 (95% CI, 0.14–0.19; *p* < 0.001)	0.89 (95% CI, 0.78–1.01; *p* = 0.074)
2nd vaccine booster	0.14 (95% CI, 0.12–0.17; *p* < 0.001)	-

OR, odds ratio; 95% CI, 95% confidence interval; COVID-19, coronavirus disease 2019; ICU, intensive care unit; SARS-CoV-2, severe acute respiratory syndrome coronavirus 2.

**Table 2 vaccines-11-00213-t002:** Efficacy the second coronavirus disease 2019 (COVID-19) vaccine booster within or after 120 days in older Italian persons (i.e., aged ≥80 years).

Endpoint	Rate (×100,000)	Odds Ratio (95% CI)
OR (95%CI) COVID-19-related hospitalizations		
≤120 days	1.14	0.67 (95% CI, 0.61–0.73; *p* < 0.001)
>120 days	1.71	
OR (95%CI) COVID-19-related ICU admissions		
≤120 days	0.02	0.53 (95% CI, 0.29–0.99; *p* = 0.045)
>120 days	0.04	
OR (95%CI) COVID-19-related deaths		
≤120 days	0.17	0.56 (95% CI, 0.45–0.69; *p* < 0.001)
>120 days	0.30	

OR, odds ratio; 95% CI, 95% confidence interval; COVID-19, coronavirus disease 2019; ICU, intensive care unit; SARS-CoV-2.

## Data Availability

Data will be available upon reasonable request to the corresponding author.

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
