# Peer review of "Efficacy of the Second COVID-19 Vaccine Booster Dose in the Elderly"

_vaccines, 2023, doi:10.3390/vaccines11020213_

Round 1
Reviewer 1 Report
Dear Authors,
congratulations to your well-written and conclusive manuscript on the efficacy of a second COVID-19 vaccine booster in older persons.
My comments are:
x) please reconsider whether the terms "elderly persons" or just "elderly" would be more appropriate than "older persons" - in both the title and in the keywords.
x) please reconsider whether the wording "seventh most lethal pandemics" is appropriate in a scientific publication (line 27)
x) the term "frail" does not refer to a person's age but to a condition; however, the categorization of individuals for this study relates only to their age, not to their general condition; the use of "frail" here should be reconsidered (line 31)
x) please indicate in the first part of the Results section (lines 56 ff) that the time that has passed since the last vaccination was not taken into account, only whether a peson was vaccinatied or boostered; in the second part of this section, you correctly mention the decline of vaccine efficacy over time
x) line 57: " ..., who received the first ...": do you mean " ... who had received the first and second vaccine boosters at November 18th, 2022"?
x) line 66: "... who did not underwent ..." -> e.g., "... who did not undergo ..."
x) line 111: please reconsider the suggestion of "regular" boostering; might this be "repeated" boostering instead?
x) Author Contributions section: Please delete the explanatory words around the actual author contributions
Best regards!
Author Response
congratulations to your well-written and conclusive manuscript on the efficacy of a second COVID-19 vaccine booster in older persons. My comments are:
- We are thankful for these valuable comments on our work. We will do our best to improve the manuscript according the suggestions.
- x) please reconsider whether the terms "elderly persons" or just "elderly" would be more appropriate than "older persons" - in both the title and in the keywords.
- ANSWER: Thanks for this comment. Title and keywords revised accordingly.
- x) please reconsider whether the wording "seventh most lethal pandemics" is appropriate in a scientific publication (line 27)
- ANSWER: Thanks for this comment. Sentence rephrased accordingly: “one of the most lethal infectious diseases”
- x) the term "frail" does not refer to a person's age but to a condition; however, the categorization of individuals for this study relates only to their age, not to their general condition; the use of "frail" here should be reconsidered (line 31)
- ANSWER: Thanks for this comment. Paper revised as suggested (i.e., “frail” has been deleted).
- x) please indicate in the first part of the Results section (lines 56 ff) that the time that has passed since the last vaccination was not taken into account, only whether a peson was vaccinatied or boostered; in the second part of this section, you correctly mention the decline of vaccine efficacy over time
- ANSWER: Thanks for this comment. Manuscript exactly revised as suggested (i.e., : this statistics does not consider the time passed since vaccination)
- x) line 57: " ..., who received the first ...": do you mean " ... who had received the first and second vaccine boosters at November 18th, 2022"?
- ANSWER: Thanks for this comment. Article revised accordingly.
- x) line 66: "... who did not underwent ..." -> e.g., "... who did not undergo ..."
- ANSWER: Thanks for this comment. Article revised accordingly.
- x) line 111: please reconsider the suggestion of "regular" boostering; might this be "repeated" boostering instead?
- ANSWER: Thanks for this comment. Text revised accordingly.
- x) Author Contributions section: Please delete the explanatory words around the actual author contributions.
- ANSWER: Thanks for this comment. Done, as suggested.
Reviewer 2 Report
This article is an interim analysis of the data from an Italian vaccination registry on the efficacy of Covid-19 vaccine boosters. The second vaccine booster showed about 10% higher efficacy than the single booster in reducing hospitalizations and deaths. However, also the efficacy of the second vaccine booster declined over time when assessed at >120 days from administration. The results are clear-cut and interesting. However, the authors should present more in detail the Italian Covid-19 surveillance program and the vaccination registry, where the data was derived (i.e. what data on vaccinations is collected to the registry, where the data comes from etc.). The English language of the article could also be improved. However, I recommend the article to be published in Vaccination after these small revisions.
Author Response
This article is an interim analysis of the data from an Italian vaccination registry on the efficacy of Covid-19 vaccine boosters. The second vaccine booster showed about 10% higher efficacy than the single booster in reducing hospitalizations and deaths. However, also the efficacy of the second vaccine booster declined over time when assessed at >120 days from administration. The results are clear-cut and interesting. However, I recommend the article to be published in Vaccination after these small revisions.
- We are thankful for these valuable comments on our work. We will do our best to improve the manuscript according the suggestions.
However, the authors should present more in detail the Italian Covid-19 surveillance program and the vaccination registry, where the data was derived (i.e. what data on vaccinations is collected to the registry, where the data comes from etc.).
- ANSWER: Very good point, thanks Information updated, as suggested, as follows: “The data published by the ISS originates from the COVID-19 integrated Italian surveillance system and are analyzed by integrating microbiological and epidemiological data provided by Regions, Autonomous Provinces and by the COVID-19 National Reference Laboratories. Specifically, all Regions and Autonomous Provinces digitally transmit daily information to the ISS concerning all individuals undergoing vaccination and those diagnosed with SARS-CoV-2 infection (i.e., all subjects with laboratory diagnosis of COVID-19, regardless of clinical signs and symptoms). The ISS has created a dedicated IT platform, interfaced with a specific database, which allows to collect data through a web interface remotely connected to the platform. All COVID-19 cases diagnosed by regional reference laboratories and other laboratories within the diagnostic network participate to the surveillance program. Information on pre-existing pathologies and on clinical conditions are also entered into the digital platform”.
The English language of the article could also be improved.
- ANSWER: Thanks for this comment. We have thoughtfully revised the text of the article to improve English language.
Reviewer 3 Report
This paper is based on the retrospective analysis of data provided by the Italian National Institute of Health, related to the COVID-19 pandemic, referring to solid endpoints such as hospitalizations, ICU admissions and deaths. The study aims to highlight the effectiveness of vaccine treatments in reducing risk in the elderly population.
The work while simple, is instructive on the objectives of the study in highlighting risk reduction relative to all end-points particularly for people who had undergone the second booster dose.
It would have been interesting for health policy purposes to assess, having the data available, whether in different population groups (e.g., 65-79, 45-64, 20-44, <20) the protective effect of the second booster dose had similar results in order to more precisely target a vaccine campaign. The conclusions appear to be in line with the premises of the study and are agreeable.
Author Response
This paper is based on the retrospective analysis of data provided by the Italian National Institute of Health, related to the COVID-19 pandemic, referring to solid endpoints such as hospitalizations, ICU admissions and deaths. The study aims to highlight the effectiveness of vaccine treatments in reducing risk in the elderly population. The work while simple, is instructive on the objectives of the study in highlighting risk reduction relative to all end-points particularly for people who had undergone the second booster dose. The conclusions appear to be in line with the premises of the study and are agreeable.
- We are thankful for these valuable comments on our work. We will do our best to improve the manuscript according the suggestions.
It would have been interesting for health policy purposes to assess, having the data available, whether in different population groups (e.g., 65-79, 45-64, 20-44, <20) the protective effect of the second booster dose had similar results in order to more precisely target a vaccine campaign.
- We ANSWER: This is a very skilful comment, which cannot unfortunately be addressed. In fact, as we specified in the introduction, “Since the administration of a second COVID-19 booster dose of mRNA-based vaccines (i.e., Pfizer/BioNTech Comirnaty and Moderna Spikevax) has been initiated in Italy from April 2022 in persons aged 80 years or older, we provide here update statistics of vaccine efficacy in this high-risk category”. Unfortunately, there is no simultaneous information on younger and elderly people that would enable to compare the efficacy of the second vaccine booster, since its administration has only commenced in people aged <60 years at the end of 2022, when different SARS-CoV-2 variants (i.e., BQ.1 and XBB.1 vs BA.5) have become endemic, thus making the comparison of second booster efficacy misleading. Moreover, the compliance to the second booster in people aged <80 years has been only around 5%, which would make the interpretation of this data very unreliable. We have included this comment at the end of the article, as possible limitation.